# A pyridinesulfonamide derivative FD268 suppresses cell proliferation and induces apoptosis via inhibiting PI3K pathway in acute myeloid leukemia

Yi Chen[1], Tianze Wu[1], Chengbin Yang[1], Mingzhu Lu[1], Zhenxia Chen[1], Mingli Deng[1], Yu Jia[1], Yongtai Yang[1], Xiaofeng Liu[1], Hongyan Wang[2], Yun Ling[1,3], Lei Lu[2]*, Yaming Zhou[1]*

1 Shanghai Key Laboratory of Molecular Catalysis and Innovative Materials, Department of Chemistry, Fudan University, Shanghai, China, 2 Institute of Metabolism and Integrative Biology, Fudan University, and Hospital of Obstetrics and Gynecology of Fudan University, Shanghai, China, 3 Zhuhai-Fudan Innovation Institute, Zhuhai, Guangdong, China

* lulei@fudan.edu.cn (LL); ymzhou@fudan.edu.cn (YZ)

## Abstract

Aberration of PI3K signaling pathway has been confirmed to be associated with several hematological malignances including acute myeloid leukemia (AML). FD268, a pyridinesulfonamide derivative characterized by the conjugation of 7-azaindole group, is a newly identified PI3K inhibitor showing high potent enzyme activity at nanomole concentration. In this study, we demonstrated that FD268 dose-dependently inhibits survival of AML cells with the efficacy superior to that of PI-103 (pan-PI3K inhibitor) and CAL-101 (selective PI3Kδ inhibitor) in the tested HL-60, MOLM-16, Mv-4-11, EOL-1 and KG-1 cell lines. Further mechanistic studies focused on HL-60 revealed that FD268 significantly inhibits the PI3K/Akt/mTOR signaling pathway, promotes the activation of pro-apoptotic protein Bad and downregulates the expression of anti-apoptotic protein Mcl-1, thus suppressing the cell proliferation and inducing caspase-3-dependent apoptosis. The bioinformatics analysis of the transcriptome sequencing data also indicated a potential involvement of the PI3K/Akt/mTOR pathway. These studies indicated that FD268 possesses high potent activity toward AML cells via inhibition of PI3K/Akt/mTOR signaling pathway, which sheds some light on the pyridinesulfonamide scaffold for further optimization and investigation.

**Data Availability Statement:** All relevant data are within the article and its Supporting information files.

## 1 Introduction

Myeloid cells originated from hematopoietic stem cells represent a diverse range of innate leukocytes which are constantly supplied to all tissues via circulation [1]. Their abnormal differentiation may cause an increase of immature malignant cells and fewer differentiated red blood cells, platelets and white blood cells, resulting into acute myeloid leukemia (AML) [2, 3]. Owing to its highly aggressive and fatal feature, targeted therapy is commonly recommended

**Funding:** The author Zhou.YM received the financial support from Zhuhai-Fudan Innovation Institute, the National Key Technologies R&D Program of China (2017YFA0205103), and Fudan-SIMM Joint Research Program (Grant No. FUSIMM20172005); author Lu.L received funding from the Commission for Science and Technology of Shanghai Municipality (20ZR1404800). The funders had no role in study design, data collection and analysis, decision to publish, or preparation of the manuscript.

**Competing interests:** The authors have declared that no competing interests exist.

as an important way to treat AML besides the bone marrow/stem cell transplant. With the increased understanding of its pathogenesis and subtype-specific pathology, various targets including vascular endothelial growth factor (VEGF) [4], isocitrate dehydrogenase (IDH) [5], Fms-like tyrosine kinase (FLT3) [6], hedgehog pathway [7, 8], ATR-CHK1 pathway [9], etc. have been revealed and specific inhibitors are thereby progressively developed [10–12]. Phosphoinositide 3-kinases (PI3Ks) are a group of heterodimeric lipid kinases regulating crucial cellular processes [13, 14]. It is observed the constitutive activation of PI3K/Akt and the mammalian target of rapamycin (mTOR) in more than 50% AML cases, which makes this signaling pathway a promising target [15–18].

Considerable PI3K/Akt/mTOR inhibitors have been constantly developed and studied either as a monotherapy or a combined therapy, providing valuable insights in AML [19–21]. BKM120, a morpholinopyrimidine derivative, has been confirmed to abrogate the activity of the PI3K/Akt/mTOR signaling, promoting cell growth arrest in AML cells [22] and impair the viability of AML cells through inducing p21-mediated $G_2$/M cell cycle arrest and reducing expression of NF-κB anti-apoptotic proteins [23]. BEZ235, an imidazole[4,5-c]quinoline derivative, has been revealed to suppress proliferation and induce apoptosis in AML cells via increasing miR-1-3p and subsequently down-regulating BAG4, EDN1 and ABCB1 [24]. The isoform-selective PI3Kδ inhibitor CAL-101, possessing a purine quinazolinone scaffold with the propeller-type conformation, has been identified to inhibit pre-rRNA synthesis and proliferation of AML cells through totally suppressing Akt phosphorylation [25]. Pan-PI3K inhibitor PI-103, belonging to pyridofuropyrimidine class, has shown antileukemic activity in AML via dephosphorylating Akt and suppressing mTORC1 activation [26]. Despite these encouraging results, AML remains a therapeutic challenge due to the intrinsically chemoresistant feature. Given the broader anti-leukemic effects associated with this pathway, the identification of novel PI3K/Akt/mTOR inhibitors and exploration of underlying mechanisms would greatly benefit to the translation to clinically optimized strategies for AML treatment.

Sulfonamide-containing compounds developed in recent years represent a large number of PI3K/Akt/mTOR inhibitors including pan- and isoform-selective inhibitors [27–31]. However, their efficacies and underlying mechanisms in AML cell lines are rarely been explored. By conjugating pyridinesulfonamide motif with 7-azaindole/azaindazole groups, we recently developed a series of PI3K inhibitors [32]. N-(5-(3-(pyridin-4-yl)-1H-pyrazolo[3,4-b]pyridin-5-yl)pyridin-3-yl) benzenesulfonamide (hereafter named as FD268, Fig 1A) is the one showing high potent enzyme activity at nanomole concentration. We herein focus on its anti-proliferation effects on AML cells. Our studies revealed that FD268 significantly inhibits their viabilities with the efficacies superior to that of PI-103 and CAL-101. Mechanistic studies revealed that FD268 suppresses cell proliferation, gets cell cycle arrests and induces apoptosis through the inhibition of PI3K activity, there by downregulating Akt phosphorylation at Ser473/Thr308 sites, inhibiting mTOR phosphorylation and downstream targets. Our studies here demonstrated the potent activity of FD268 and the underlying therapeutic mechanism, suggesting a novel pyridinesulfonamide scaffold worthy of further investigation.

## 2 Materials and methods

### 2.1 Chemicals

The compound FD268 was synthesized in our laboratory. Pan-PI3K inhibitor PI-103 and isoform-specific PI3K inhibitor CAL-101 were purchased from Selleckchem (Houston, Texas, USA). Stock solutions of all the inhibitors were prepared in dimethyl sulfoxide (DMSO), and 10 μL aliquots were frozen at -80˚C.

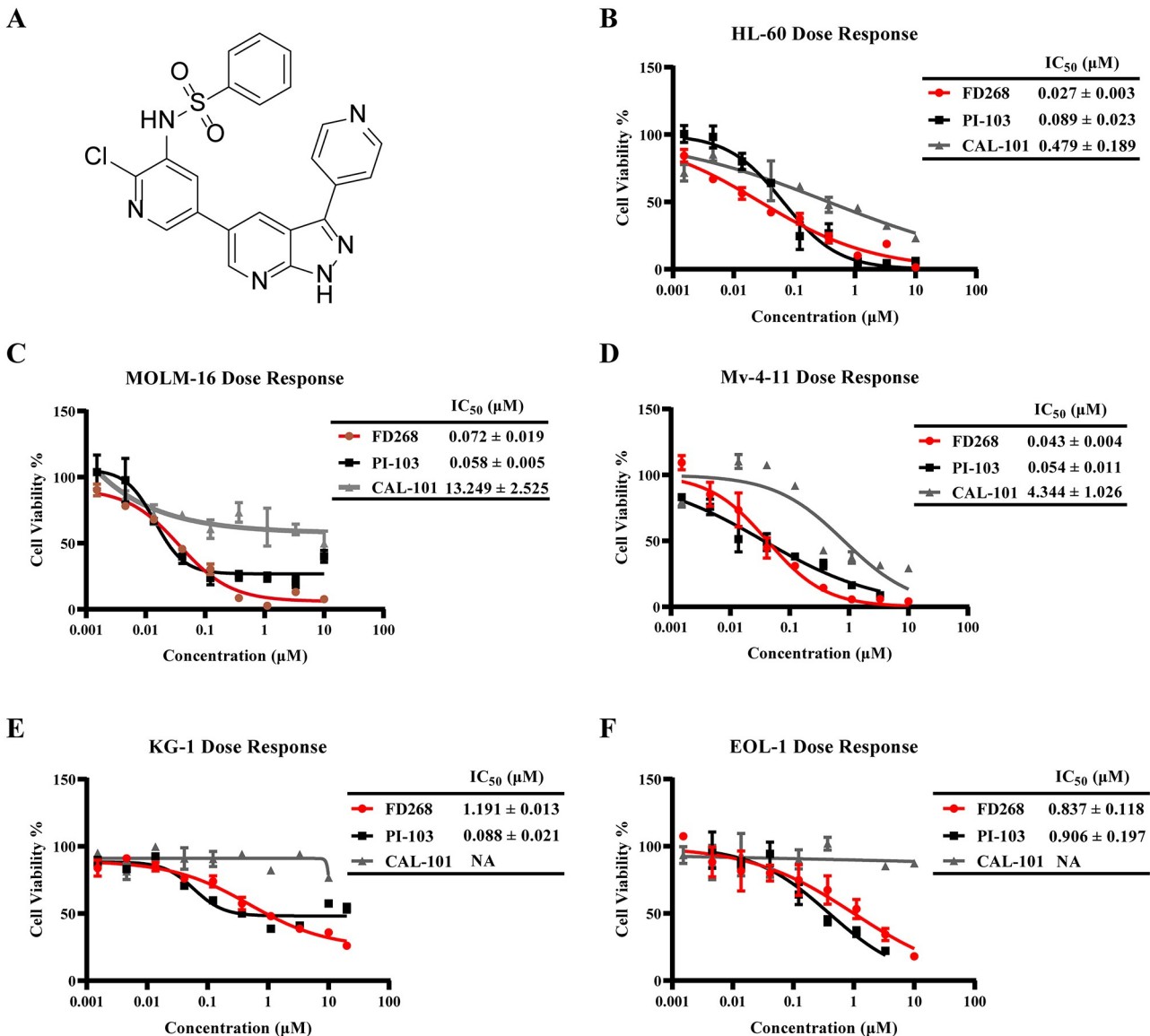

**Fig 1. FD268 inhibits cell viability of AML cell lines.** (A) Structure of the compound FD268. (B-F) Icreasing concentrations of FD268, PI-103, and CAL-101 were used to treat HL-60, MOLM-16, Mv-4-11, KG-1 and EOL-1 cells for 48 h, and the antiproliferative activity of FD268 against five AML cell lines was determined by the CCK-8 assay in triplicate. Growth inhibitory curves and the IC$_{50}$ values of FD268, PI-103, and CAL-101 in AML cells were also analyzed. The values were shown as the mean ± standard from three independent experiments. NA means no real available value.

## 2.2 Cell culture and treatment

Human leukemic cell lines, MOLM-16 (established from the peripheral blood of a 20-year-old man with acute myeloid leukemia AML) [33], Mv-4-11 (established from the peripheral blood of a 20-year-old man with acute myeloid leukemia AML) [34], EOL-1 (derived from the peripheral blood of a 33-year-old man with hypereosinophilia case terminating in blastic transformation.) [35] and KG-1 (obtained from a 59-year-old Caucasian male with erythroleukemia that evolved into acute myelogenous leukemia) [36], were purchased from Shanghai Bogoo Biotechnology Co., Ltd (Shanghai, China), and the HL-60 (obtained by leukopheresis from a 36-vear-old Caucasian female with acute promvelocytic leukemia) [37] cell line was

from the American Type Culture Collection (ATCC, Manassas, VA, USA). The cells were grown in RPMI-1640 media containing 10% fetal bovine serum (Gibco Life Technologies, Gaithersburg, MD, USA) and 1% penicillin-streptomycin solution (Gibco Life Technologies, Gaithersburg, MD, USA) at 37˚C in a humidified atmosphere containing 5% $CO_2$.

## 2.3 Cell viability assays

Cell viability assays were investigated by the Cell Counting Kit-8 (CCK-8; Vamzyme, Nanjing, China). Briefly, cells were seeded in triplicates at a density of $1 \times 10^4$ cells per well on 96-well plates and allowed to adhere overnight. Cells were incubated with different concentrations of PI-103, CAL-101 and FD268 for 48 h. The controls were incubated with 0.1% DMSO. CCK-8 solution was added to each well and further incubated for 2 h at 37˚C. The absorbance of optical density was measured at 450 nm using a multi-well spectrophotometer (BioTek Instruments, Inc., mit Hauptsitz in Winooski, VT, USA). The inhibitory rate was calculated using the following formula: 1 –(experimental group OD value/control group OD value) ×100% and the $IC_{50}$ value was calculated by fitting the data to a logistic curve using GraphPad Prism 7 software (GraphPad Software, Inc., San Diego, CA, USA). At least three independent experiments were carried out in each cell line.

## 2.4 Cell cycle and apoptosis analysis

Cells were seeded in triplicates at a density of $5 \times 10^5$ cells (cell cycle assays) and $4 \times 10^5$ cells (apoptosis assays) per well in 6-well plates and treated with FD268 (0.125, 0.25 and 0.5μM) or DMSO control for 24 h and 48 h, respectively. Cells were harvested after treatment and washed with PBS. For cell-cycle distribution analysis, cells were fixed in ice-cold 70% ethanol, stained by the Propidium Iodide (PI) staining solution (Thermo Fisher Scientific, Inc., Waltham, MA, USA) and then incubated at room-temperature in dark for 15 min. For cell apoptosis assays, cells were stained using FITC Annexin V/propidium iodide (PI) Apoptosis Detection Kit (Vamzyme, Nanjing, China) according to the manufacturer's protocol. The cell cycle analysis and early apoptosis analysis were conducted using the Flow Cytometer system (Beckman Coulter, Brea, CA, USA). The percentage of cells in each phase was calculated using ModFit LT software (Verity Software House, Inc., ME, USA), the cell apoptosis data analysis was performed using FlowJo software (BD Biosciences, San Jose, CA, USA). At least three independent experiments were carried out in cell apoptosis assay.

## 2.5 Western blot analysis

Cells were seeded in triplicates at a density of $1 \times 10^6$ cells per well in 6-well plates and treated with FD268 (0.063, 0.125, 0.25, 0.5 and 1 μM, respectively) or DMSO control for 24 h and then lysed with ice-cold radioimmunoprecipitation (RIPA) buffer (Thermo Fisher Scientific, Inc, Waltham, MA, USA) containing protease and phosphatase inhibitor cocktail tablet (Roche, Mannheim, Germany). Total protein concentrations were measured using a BCA protein assay kit (Thermo Fisher Scientific Inc., Waltham, MA, USA). Equal amounts of protein were separated on SDS-polyacrylamide gels (10, 12%), and electro-transferred to PVDF membranes. Nonspecific protein binding was blocked with 5% bovine serum albumin (BSA) for 1.5 h at room temperature and probed with the primary antibodies overnight at a 1:1000 dilution at 4˚C. The primary antibodies: anti-β-actin polyclonal antibody (Cat#10494-1-AP), Phospho-Akt (Ser473) (Cat#4060), Phospho-Akt (T308) (Cat#13038), Akt (Cat#4685), Phospho-mTOR (Ser2448) (Cat#5536), mTOR (Cat#2983), Phospho-4EBP1 (Ser65) (Cat#9451), 4EBP-1 (Cat#9644), Phospho-p70 S6 Kinase (Thr389) (Cat#9205), caspase-3 (Cat#14220), cleaved caspase-3 (Cat#9661), cleaved PARP (Cat#5625), Bad (Cat#9239), Bax (Cat#5023), Bcl-2

(Cat#15071), Mcl-1 (Cat#94296), and Bcl-XL (Cat#2764) were from Cell Signaling Technology (Beverly, MA, USA). Blots were incubated with 1:3000 dilution sheep anti-rabbit IgG (Cat#7074, Cell Signaling Technology Inc., Beverly, MA, USA) at room temperature for 1 h, and immunoreactive protein bands were visualized using Pierce ECL Western blotting substrate (Thermo Fisher Scientific Inc., Waltham, MA, USA). The blot density was determined by FUSION FX software (Vilber Lourmat Inc., France). At least two independent experiments were carried out in western blot assay.

## 2.6 Immunofluorescence

HL-60 cells were seeded in triplicates at a density of $1 \times 10^6$ cells per well in 6-well plates and treated with FD268 (0.125 and 0.5 μM, respectively) or DMSO control for 24 h. Immunofluorescence staining was performed according to the standard operating procedure. Cells were washed with PBS and fixed in pre-cooled methanol for 30 min. Methanol was removed and cells were washed twice with PBS, followed by permeation with 0.2% Triton X-100 for 30 min at room temperature. Cells were washed twice with PBS and blocked with 2% BSA for 1 h at 37˚C. Cells were then incubated with primary antibody against p-Akt (Ser473) (1:200, Cell Signaling Technology Inc., Beverly, MA, USA) in 1% BSA at 4˚C overnight. Cells were washed twice with PBS and then incubated with secondary antibodies CoraLite594-conjugated Goat-Anti-Rabbit IgG(H+L) (1:200, Cell Signaling Technology Inc., Beverly, MA, USA) in PBS for 1 h at 37˚C. Cells were washed twice with PBS, and nuclei were stained with DAPI for 3 min at room temperature. Cells were washed twice with PBS and visualized. Images were obtained by a fluorescence microscopy.

## 2.7 RNA extraction and real-time RT-PCR

Cells were seeded in triplicates at a density of $1 \times 10^6$ cells per well in 6-well plates and treated with FD268 or DMSO as control for 24 h. Total RNA was prepared using the TRIzol reagent (Life Technologies, Grand Island, NY, USA) according to the manufacturer's instructions. Briefly, one microgram of isolated total RNA was reverse transcribed using SuperScript III Reverse Transcriptase kit (Vamzyme, Nanjing, China) to complementary DNA (cDNA). Expression of relative genes of FD268-treated cells was measured by SYBR green based real-time quantitative PCR using Fast SYBR Green Master Mix (Vamzyme, Nanjing, China) at the StepOne RealTime PCR System. All real-time PCR reactions were performed in triplicate. A melting curve analysis was applied to verify the specificity of the products, and the values for relative quantification were calculated based on $2^{-\Delta\Delta Ct}$ relative expression formula and normalized against the geometric average expression HPRT1 housekeeping genes. Nucleotide sequences of the primers (Life Technologies) used for qRT-PCR were listed in S1 Table.

## 2.8 RNA sequencing and Kyoto Encyclopedia of Genes and Genomes pathway analysis

HL-60 cells were seeded in triplicates at a density of $1 \times 10^6$ cells per well in 6-well plates and treated with FD268 (0.125, 0.5 μM) or DMSO for 24 h. Cells were harvested and the total RNA was extracted for later use. 1000 ng high-quality digested RNA was employed using the VAHTS mRNA-seq v2 Library Prep kit for Illumina (Vamzyme, Nanjing, China), and sequencing libraries were created according to the manufacturer's protocol. Briefly, first- and second-strand cDNA were synthesized using random hexamer primers. The End-It DNA End Repair kit was used to repair the cDNA fragments. An "A" nucleotide was added to the 30 ends of the fragments, followed by adapter ligation. The ligated cDNA products were subjected to PCR amplification. The library quality was determined using a Bioanalyzer 2100 (Agilent,

Santa Clara, CA, USA). The Illumina Nova600 platform (Illumina, San Diego, CA, USA) was used for RNA-seq. The quality of RNA Sequencing (RNA-seq) data was evaluated using FastQC. Kyoto Encyclopedia of Genes and Genomes (KEGG) pathway analysis was applied to identify the significant pathway of the differentially expressed mRNAs. Fisher's exact test was used to identify the significant results, and the false discovery rate (FDR) was applied to correct the p-values (p value < 0.05; fold change > 1.5).

## 2.9 Statistical analysis

All experiments were analyzed using one-way ANOVA in Graphpad Prism 8.0. (GraphPad Software, Inc., La Jolla, CA, USA). Experiments were presented as the means ± standard error in three independent experiments. Statistical analysis was performed using One-way ANOVA to determine the significance between groups. Differences were considered statistically significant when the P value was less than 0.05 (*).

# 3 Results

## 3.1 FD268 exhibits antiproliferative activity against AML cells

Conjugation of pyridinesulfonamide motif with 7-azaindazole group led to the screen out of FD268. It has been identified as a potent pan-PI3K inhibitor with the IC50 value of ~1, ~4, ~7 and ~0.7 nM for PI3K α, β, γ and δ isomer respectively [32]. Since the identified association of the poor prognosis and chemoresistance in AMLs with the PI3K/Akt/mTOR activation, we herein investigated the antiproliferative activity of FD268 against five hematological cancer cell lines: HL-60, MOLM-16, Mv-4-11, KG-1 and EOL-1. As shown in Fig 1, FD268 exhibited a clear dose-dependent inhibitory activity with the 50% inhibitory concentration (IC50) values of 0.027 ± 0.003, 0.072 ± 0.019, 0.043 ± 0.004, 1.191 ± 0.013 and 0.837 ± 0.118 μM respectively. The inhibition activity was comparable to that of pan-PI3K inhibitor PI-103, among which a ~40 fold enhancement was observed in HL-60. On the other hand, the inhibitory activity of FD268 was superior to that of clinical available PI3Kδ selective inhibitor CAL-101 in all tested lines (0.479 ± 0.189, 13.25 ± 2.52, 4.43 ± 1.01μM for HL-60, MOLM-16 and Mv-4-11), especially in MOLM-16 with a ~100 fold enhancement. These results demonstrated that FD268 possessed strong antiproliferative activity against the tested AML cells, especially in HL-60 and MOLM-16.

## 3.2 FD268 induces cell cycle arrest and apoptosis in AML cells

Given the high potent activity of FD268 against HL-60 and MOLM-16 as well as their corresponding morphological changes (observable cell shrinkage, cytoplasmic condensation and nuclear fragmentation) (S1 Fig.), the cell cycle and apoptosis were consequently analyzed. As shown in Fig 2, the cell population distributed in $G_0/G_1$, S and $G_2/M$ phase was 35.5 ± 3.7%, 45.8 ± 2.2% and 18.8 ± 3.7% respectively for the untreated HL-60. When incubated with FD268, the proportion of $G_0/G_1$ phase cells increased to 52.9 ± 2.4%, meanwhile S and $G_2/M$ phase cells decreased to 34.2 ± 4.4% and 12.9 ± 3.6% respectively, suggesting that the cells were arrested in the $G_1$ phase. In contrast, treatment of FD268 induced the increase of $G_2/M$ phase cells from 12.3 ± 2.1% to 24.1 ± 6.8% in MOLM-16, indicating the $G_2$ phase cycle arrest. The cell-line-dependent cell-cycle arrest and the observable cell morphological changes raised our interest whether FD268 exerted distinct modes of cell death. Thus, FD268-treated HL-60 and MOLM-16 were analyzed by flow cytometry with annexin V/propidium iodide double staining. The flow cytometric data showed that the total proportion of AnnexinV-FITC positive and AnnexinV-FITC/PI positive cells dose-dependently increased from 1.8 ± 0.3% (control

**A**

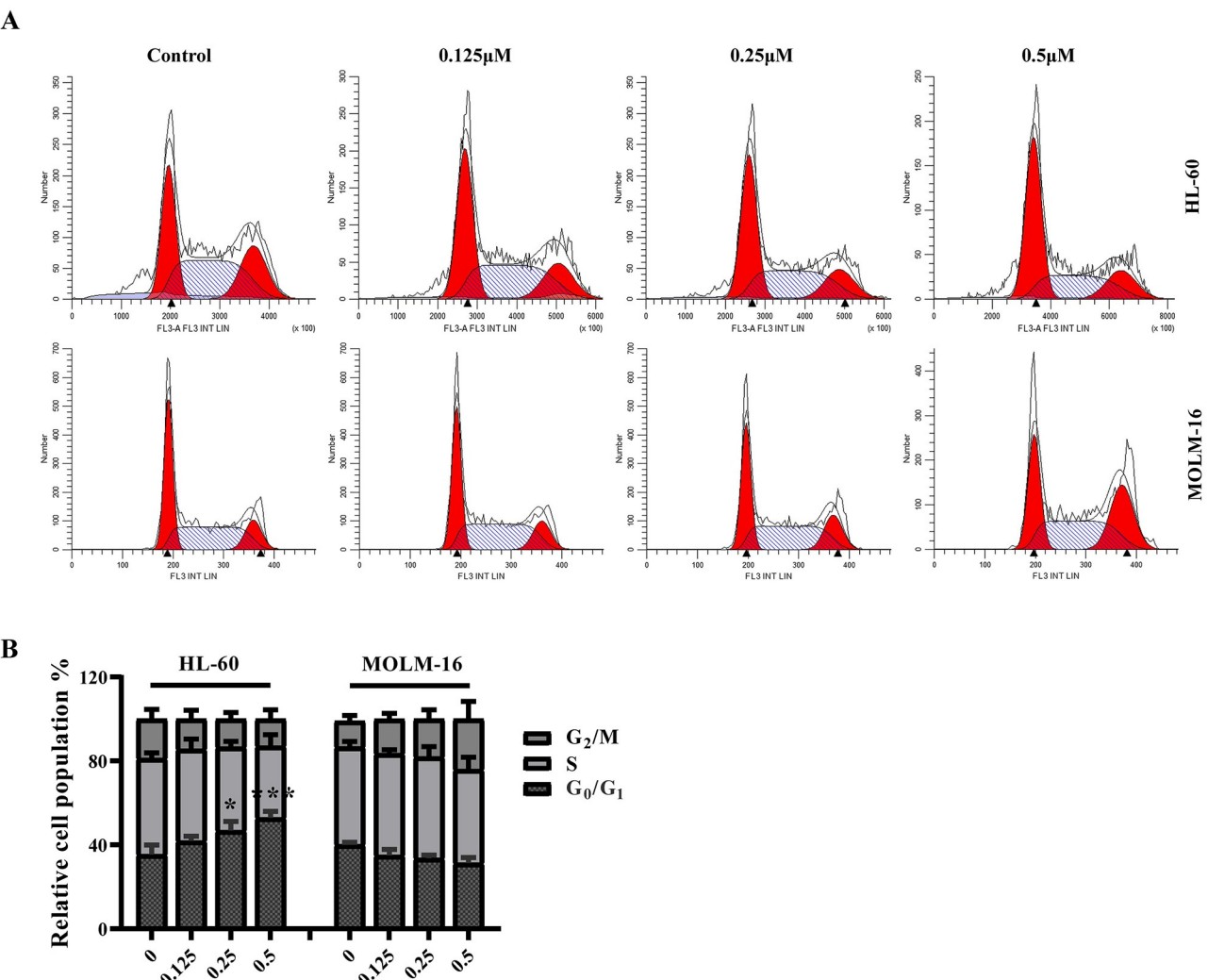

**B**

**Fig 2. FD268 induces cell cycle arrest in HL-60 and MOLM-16 cells.** (A) Induction of cell cycle arrest by FD268 at the increasing concentrations for 24 h in HL-60 and MOLM-16. Cell cycle phase analysis was performed by flow cytometry. (B) Quantitation of the $G_0/G_1$, S, and $G_2$ phase cells of HL-60 and MOLM-16. The data shown are the mean ± standard from two independent experiments. Statistical differences compared to control group were analyzed with one-way ANOVA. Statistical significance is represented with asterisks *$P < 0.05$ and ***$P < 0.001$.

group) to 28.8 ± 4.3% (0.5μM-FD268-treated groups) in HL-60 and from 2.5 ± 0.1% to 10.0 ± 2.1% in MOLM-16 (Fig 3A and 3B). Considering the cascade processes of caspase-3 and its substrate poly (ADPribose) polymerase (PARP) in inducing cell programmed death, the cleavages of these two proteins were consequently examined by western blot assay, which showed a dose dependent increase of both caspase-3 and PARP cleavage. (Fig 3C). These results demonstrated that FD268 induces apoptosis of the tested two cell lines.

To take mechanistic insights into the cell-line-dependent cell-cycle arrest and apoptosis, the expression of cyclin-dependent kinase 2 (CDK2) and CDKs inhibitor (p21Cip1, p27Kip1) mRNA were firstly measured. Treatment of 0.25 μM FD268 resulted into a decrease in the expression level of p21Cip1, p27Kip1 and CDK2 in HL-60, but an increase of p27Kip1 and CDK2 in MOLM-16 (Fig 4A and 4B). Since that CDK2 mRNA appeared late in G1 or in early S phase, the differences in the expression may lead to the cell-line-dependent cell-cycle arrest. Nevertheless, on protein level, CDK2 showed decreases to 40 and 60% of the initial level in

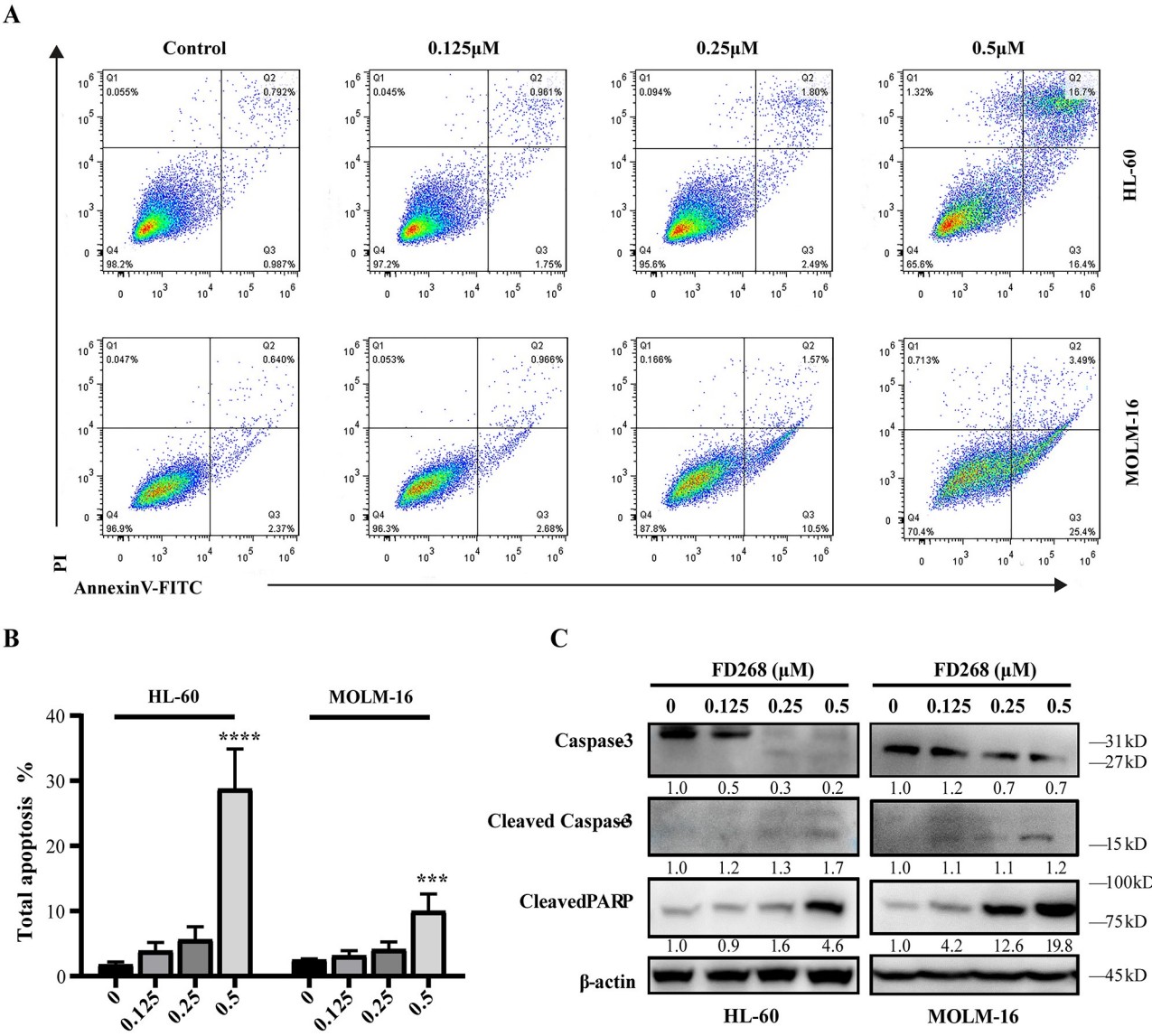

**Fig 3. FD268 induces cell apoptosis in HL-60 and MOLM-16 cells.** (A) Induction of cell apoptosis by FD268 at the increasing concentrations for 48h. Apoptotic cells were quantified by flow cytometry after staining with PI and FITC-Annexin V. (B) The summary of populations stained with only Annexin V (early apoptosis) and with both Annexin V and PI (late apoptosis). The data shown are the mean ± standard from three independent experiments. Statistical differences compared to control group were analyzed with one-way ANOVA. Statistical significance is represented with asterisks ***P < 0.001, ****P < 0.0001. (C) Western blot was applied to detect cleaved PARP, cleaved caspase-3 and caspase-3 as indicators of apoptotic cell death. Intensity of the bands was measured, and the fold change of the intensity was compared with that of the control and indicated below the bands. β-actin was used as a loading control.

HL-60 and MOLM-16 by FD268 (Fig 4C). To investigate the underlying mechanism of the cell apoptosis, the expression of Bcl-2 family members, Bad, Bcl-2 and Mcl-1, were consequently measured on the transcriptional and protein level for the tested two cell lines respectively. At the transcriptional level, the anti-apoptosis gene Bcl-2 and Mcl-1 were both dose-dependently downregulated in HL-60, while the pro-apoptotic gene Bad was dose-independent (Fig 5A). In contrast, the gene Bad, Bcl-2 and Mcl-1 was respectively upregulated in MOLM-16 induced by FD268 (Fig 5B). At protein level, FD268 induced a 50% increase in the expression level of pro-apoptotic protein Bad, meanwhile inhibited the anti-apoptotic protein Mcl-1 in both HL-60

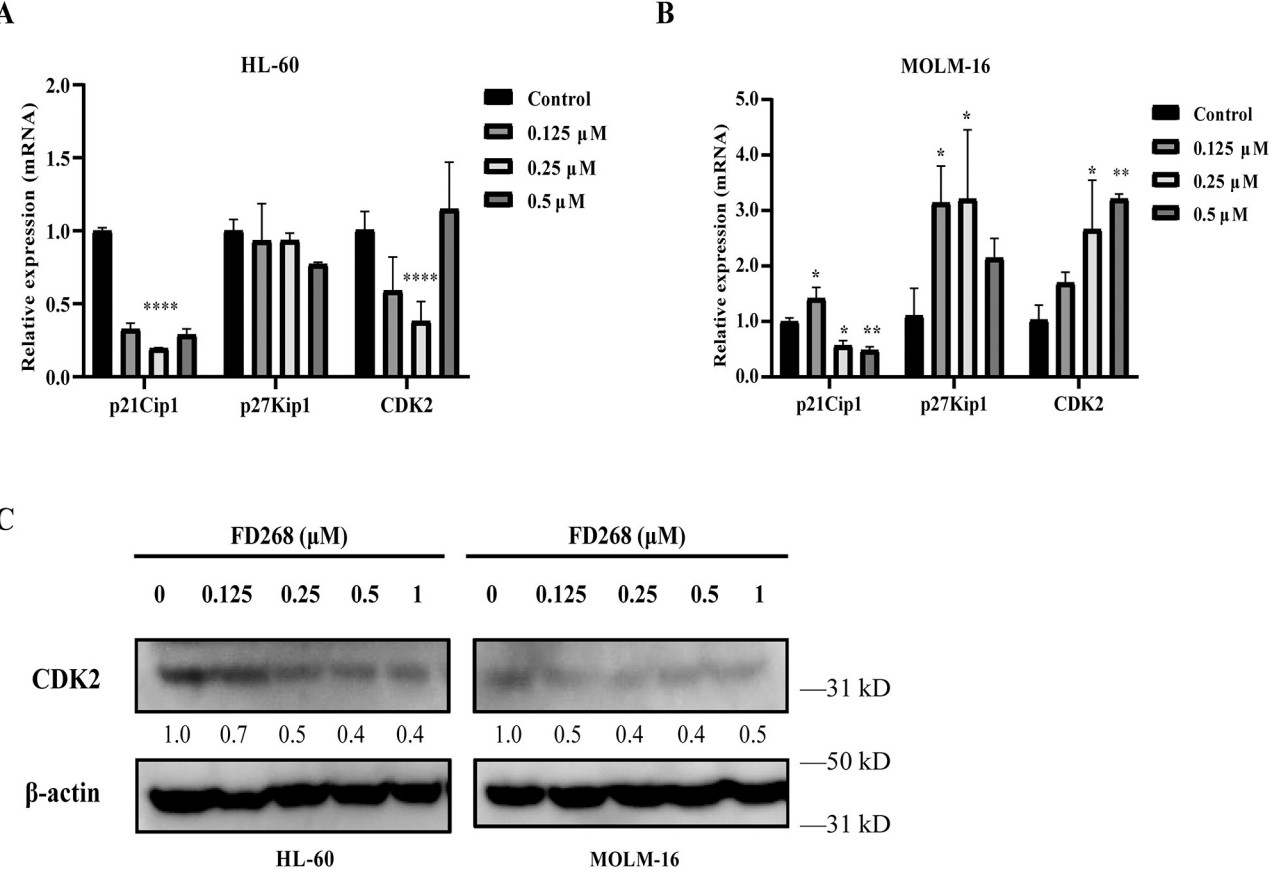

**Fig 4. FD268 regulates the expressions of cell cycle related protein in HL-60 and MOLM-16 cells.** Cells were treated with increasing concentrations of FD268 for 24 h. Transcriptional levels of cell cycle related genes p21Cip1, p27Kip1 and CDK2 were determined using qRT-PCR in HL-60 (A) and MOLM-16 (B) cells. Two-way ANOVA followed by Tukey's multiple comparisons test. Statistical significance is represented with asterisks *P < 0.05, **P < 0.01, ****P < 0.0001. (C) The levels of apoptosis-related protein CDK2 in control and FD268-treated cells were determined by western blot. Intensity of the bands was measured, and the fold change of the intensity was compared with that of the control and indicated below the bands. β-actin was used as a loading control.

and MOLM-16 dose-dependently (Fig 5C). While, the protein levels of other markers of apoptosis, Bcl-2, Bax and Bcl-XL, did not change significantly in FD268-treated cells. These results demonstrated that FD268 could induce the cell cycle arrest and apoptosis in HL-60 and MOLM-16 while the associated inhibition mechanisms are slightly different.

### 3.3 FD268 inhibits the activation of PI3K/Akt/mTOR signaling in AML cells

Considering the differences of cell cycle arrest and apoptosis at gene level, we examined the phosphorylation status of AKT, a marker for cellular PI3K activity, and downstream substrates of PI3K signaling in HL-60 and MOLM-16 induced by FD268. As shown in Fig 6, FD268 dose-dependently inhibited the expression of phosphorylated Akt (Ser473 and Thr308) after 24 h. The inhibition of phosphor-Akt (Ser473) expression was also confirmed by the immunofluorescence assay (S2 Fig.). These results indicated that FD268 did effectively inhibit PI3K signaling cascades in HL-60 and MOLM-16. We further determined the effects of FD268 on mTOR activity downstream the PI3K/Akt signaling. Western blot assay showed that FD268

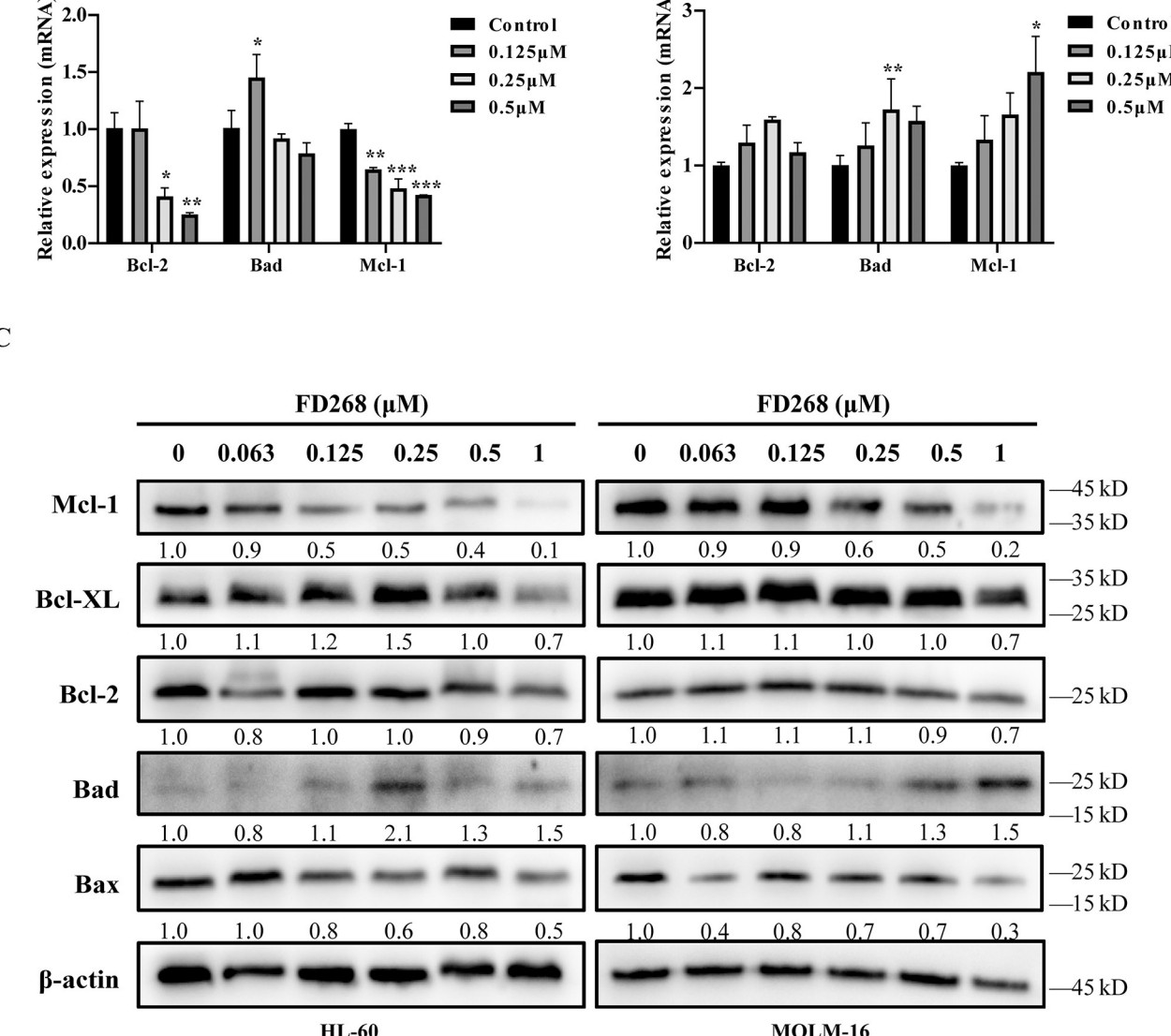

**Fig 5. FD268 regulates Bcl-2 family apoptotic proteins in HL-60 and MOLM-16 cells.** Cells were treated with increasing concentrations of FD268 for 24 h. Transcriptional levels of apoptosis-related genes Bcl-2, Bad and Mcl-1 were determined using qRT-PCR in HL-60 (A) and MOLM-16 (B) cells. Two-way ANOVA followed by Tukey's multiple comparisons test. Statistical significance is represented with asterisks $^{*}$P < 0.05, $^{**}$P < 0.01, $^{***}$P < 0.001. (C) The levels of apoptosis-related proteins including Bad, Mcl-1, Bcl-2, Bax, Bcl-XL, and in control and FD268-treated cells were determined by western blot. Intensity of the bands was measured, and the fold change of the intensity was compared with that of the control and indicated below the bands. β-actin was used as a loading control.

dose-dependently inhibited the phosphorylation of mTOR at Ser2448 site, showing a 50% and 80% decrease of the initial level in HL-60 and MOLM-16 induced by 1 μM FD268 after 24 h respectively. Furthermore, FD268 almost fully inhibited the expression of mTOR downstream factors p-4E-BP1 (Ser65) and p-p70S6K (Thr389) at 1 μM concentration. These results revealed that FD268 exerted the antiproliferation and the induction of apoptosis by inhibition of PI3K/Akt/mTOR signaling pathway in HL-60 and MOLM-16.

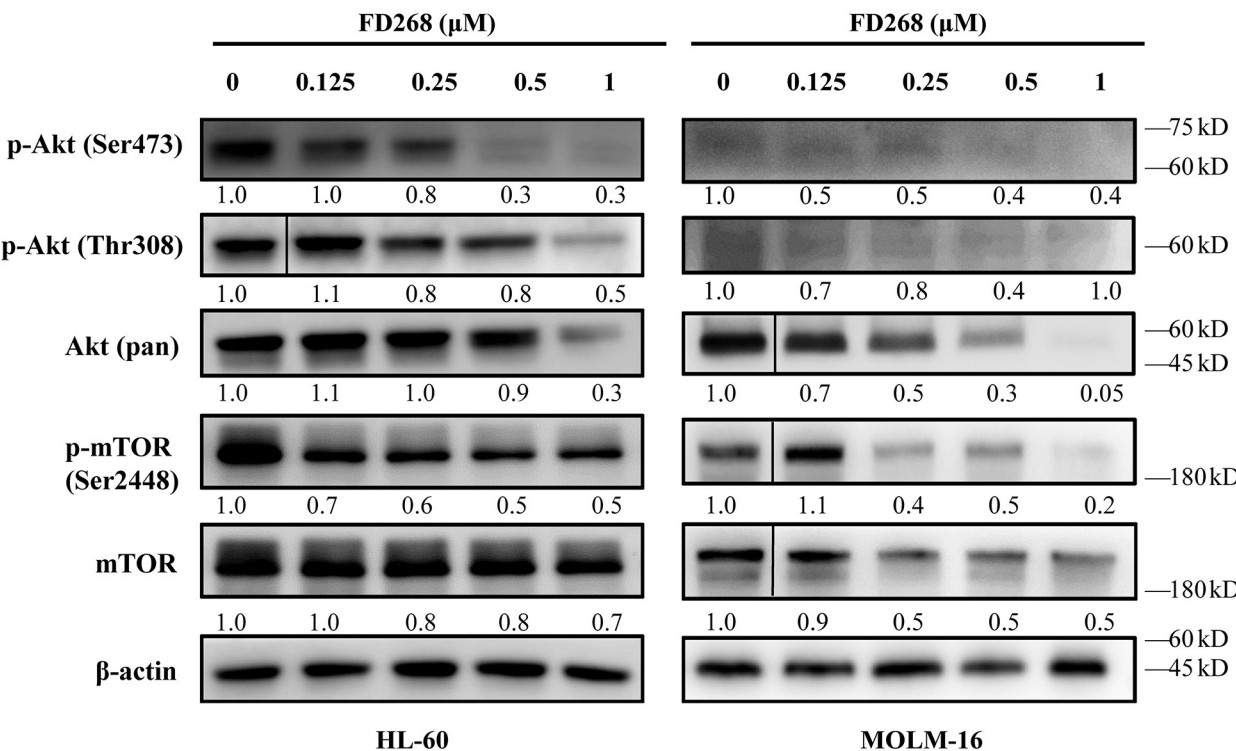

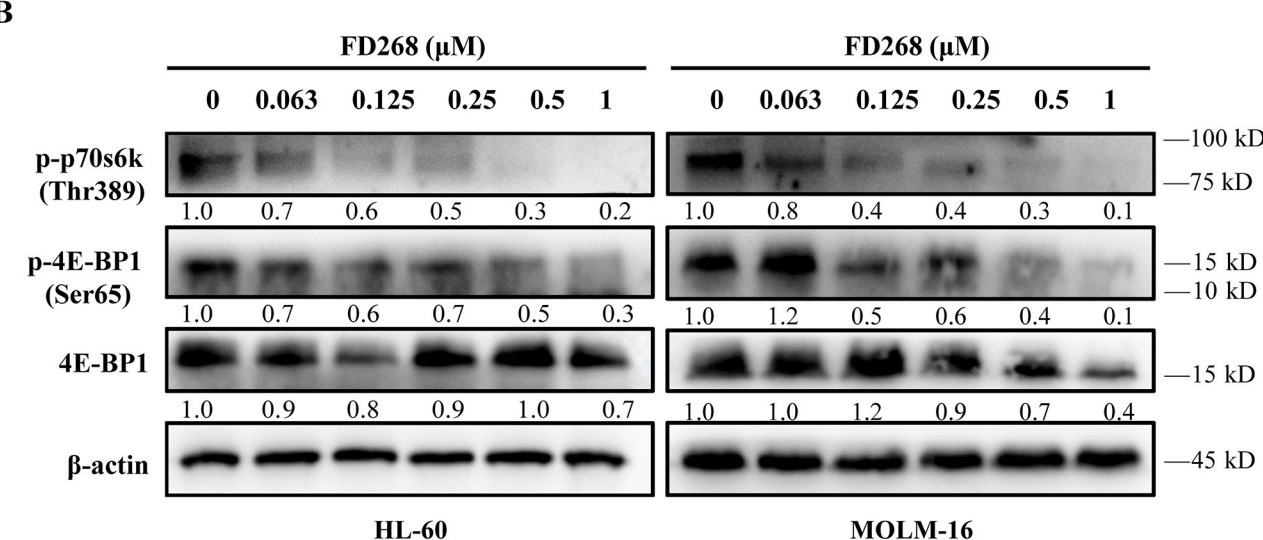

**Fig 6. FD268 inhibits the PI3K/Akt/mTOR signaling in HL-60 and MOLM-16 cells.** HL-60 and MOLM-16 cells were treated with FD268 at increasing concentrations for 24h and harvested to measure the levels of PI3K pathway proteins by western blot. (A) AKT, p-AKT (Ser473), p-AKT (Thr308), mTOR, and p-mTOR (Ser2448) was detected. (B) The level of phosphorylated mTOR target proteins: p-4E-BP1 and p70 ribosomal protein S6 kinase (p-p70S6K) was detected by western blot. FD268 concentration-dependently inhibited the phosphorylation of AKT, mTOR and downstream targets of PI3K pathway. Intensity of the bands was measured, and the fold change of the intensity was compared with that of the control and indicated below the bands. β-actin was used as a loading control.

### 3.4 FD268 regulates cell survival related genes and pathways in AML cells

To further provide a detailed and unbiased view of the complex and dynamic nature of the transcriptome, RNA-seq analyses were performed on FD268-treated HL-60 cells (0.125 and 0.5 μM respectively for 24 h). All data were analyzed after quality screening, and DESeq2 software was used for screening differentially expressed genes between control and FD268-treated groups (adjusted p-value < 0.05; fold change > 1). There were 194 upregulated and 148 downregulated genes observed in 0.125 μM FD268 treated group. With the increase of concentration to 0.5 μM, the numbers of upregulated and downregulated genes also increased to 655 and 240, respectively. A heat map (Fig 7A) and volcano plots (S3 Fig) of the differentially

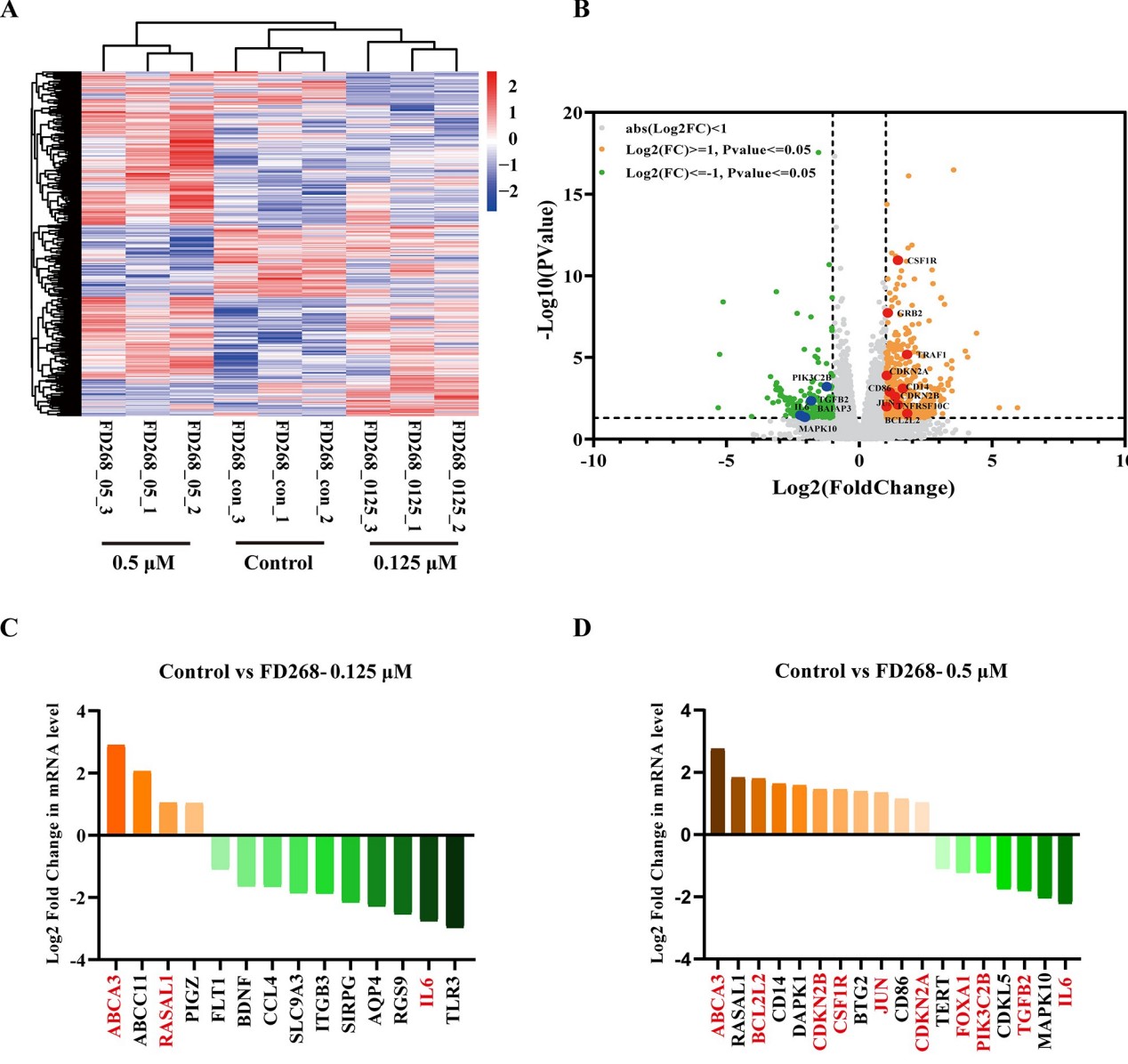

**Fig 7. FD268 induces differential regulation of gene expression in HL-60 cells.** (A) HL-60 cells were treated with 0.125 and 0.5 μM FD268 for 24 h at the indicated concentrations and collected for RNAseq analysis. Hierarchical clustering shows deferentially expressed mRNA patterns between the control and FD268-treated groups. (B) Volcano plot of deferentially expressed mRNAs of 0.5 μM FD268-treated group. A total of 895 deferentially expressed mRNAs were identified, including 655 upregulated and 240 downregulated mRNAs. (C) (D) Crucial regulators of cell proliferation screened from each FD268-treated group in HL-60 cells. DESeq2 was used to screen differentially expressed genes among different groups (|log2FC| ≥1.5 and P value≥0.05).

expressed genes (DEGs) showed a clear distinction between control group and FD268-treated groups. Among the FD268-treated groups, BCL2L2, JUN, TARF11 and TNFRSF10C were related to apoptosis; CDKN2A, CDKN2B and TGFB2 were related to cell cycle, ABCA3 and ABCC11 were related to ATP binding. It is worth to mention that a few genes with a role in acute myeloid leukemia: GRB2, CSFIR, MARCKS and PDGFB were upregulated; FLT1, IL6, PIK3C2B, BALAP3 and MAPK10 were downregulated (Fig 7B–7D).

GO analysis was used to identify the functional categories of these deferentially expressed genes. A total of 1669 GO terms in the category of biological processes were enriched at an FDR threshold of $< 0.05$. The top eight significantly enriched biological processes were inflammatory response, defense response, response to wounding, regulation of response to stimulus, secretion, secretion by cell, cytokine production and immune response (S4 Fig and S2 Table). Pathways were annotated and categorized in KEGG database for the identified differential target genes (S3 Table). Top 6 pathways were therefore isolated and shown in the statistical diagram (Fig 8A). In the 0.125 μM FD268 treated group, downregulated genes were mainly enriched in Toll-like receptor signaling pathway, PI3K-Akt signaling pathway and cytosolic DNA-sensing pathway. PI3K-Akt signaling pathway was significantly enriched with extremely low p-value, indicating that FD268 significantly altered PI3K-Akt pathway genes in human leukemia HL-60 cells which was consistent with the above western blot results.

Furthermore, a set of genes were identified by overlapping the deferentially expressed target genes of 0.125 μM FD268-treated and 0.5 μM FD268-treated group (S4 Table). ABCA3, IL6, RHOB, NOXA1, C1RL, LRRC25, RASAL1 and SULF2 were significantly regulated by FD268 dose-dependently (Fig 8B and 8C). Among these targets, ATP transmembrane transporter subfamily protein ABCA3 and NADPH oxidase activator NOXA1 are related to apoptosis; LRRC25 promotes autophagy and induces the inhibition of NF-κB signaling pathway [38]; cytokine interleukin 6 is also closely related to PI3K pathway and cell differentiation [39, 40]. The alternation of these target genes was verified by RT-qPCR (S5 Fig). Therefore, it is reasonable to conclude that FD268 interrupt cell proliferation via regulating a bunch of crucial PI3K/Akt related genes and pathways.

## 4 Discussion

Inhibition of PI3K is believed to induce apoptosis in cancer cells [41]. Developed in parallel to those of PI-103 and CAL-101 analogous [42, 43], pyridinesulfonamide derivatives are another class of PI3K inhibitors generally characterized by occupying the affinity pocket of PI3K with pyridinesulfonamide motif leaving the hinge region and hydrophobic region of PI3K to the conjugated groups. FD268 is the one conjugated with 7-azaindole motif showing the potent activities against class I PI3K [32]. Owing the verified hyperactivation of PI3K/Akt/mTOR in AML, targeting PI3K is therefore of great therapeutic interest for the treatment of AML. Although the pyridinesulfonamide derivatives represented a large number of PI3K inhibitors developed recently, their efficacies as well as plausible mechanistic paths in AML are seldom explored. Thus, whether and how FD268 suppresses the proliferation and promotes apoptosis induction through inhibition of the PI3K network activity in AML cell lines received our great interest.

We examined for the first time the in vitro activity of FD268 on a panel of human AML cell lines including HL-60, MOLM-16, Mv-4-11 and KG-1, which are characterized by different PI3K/Akt/mTOR pathway activation patterns. Our studies demonstrated that FD268 significantly suppresses cell proliferation in a dose-dependent manner for the tested cell lines with the potent activities superior to that of PI-103 and CAL-101. Given its high anti-proliferation activity on HL-60 and MOLM-16, FD268-induced cell cycle arrest and apoptosis were

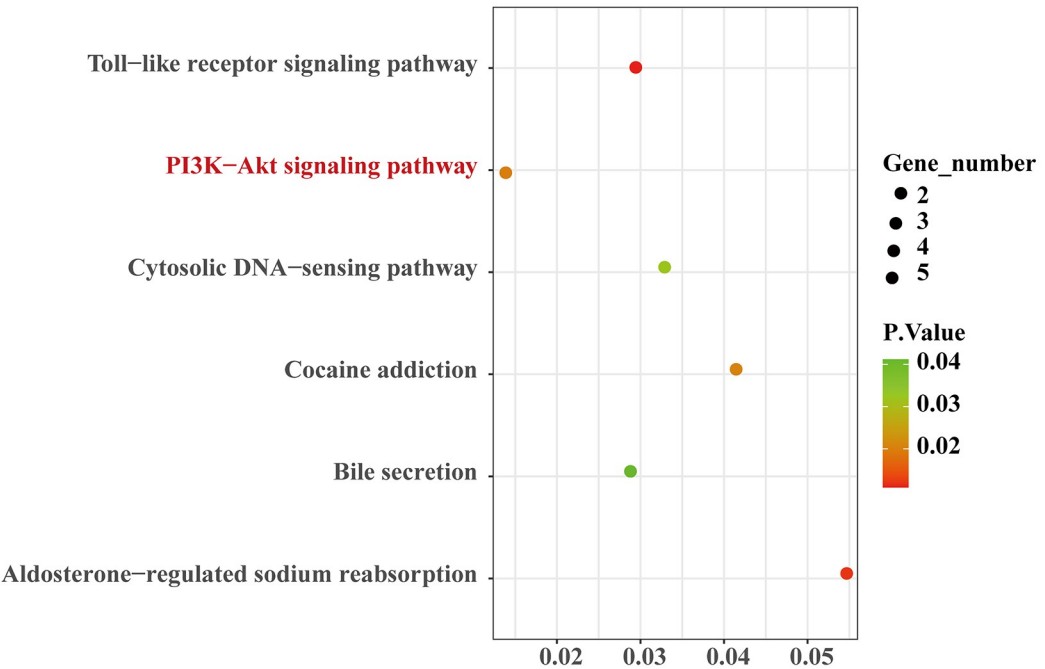

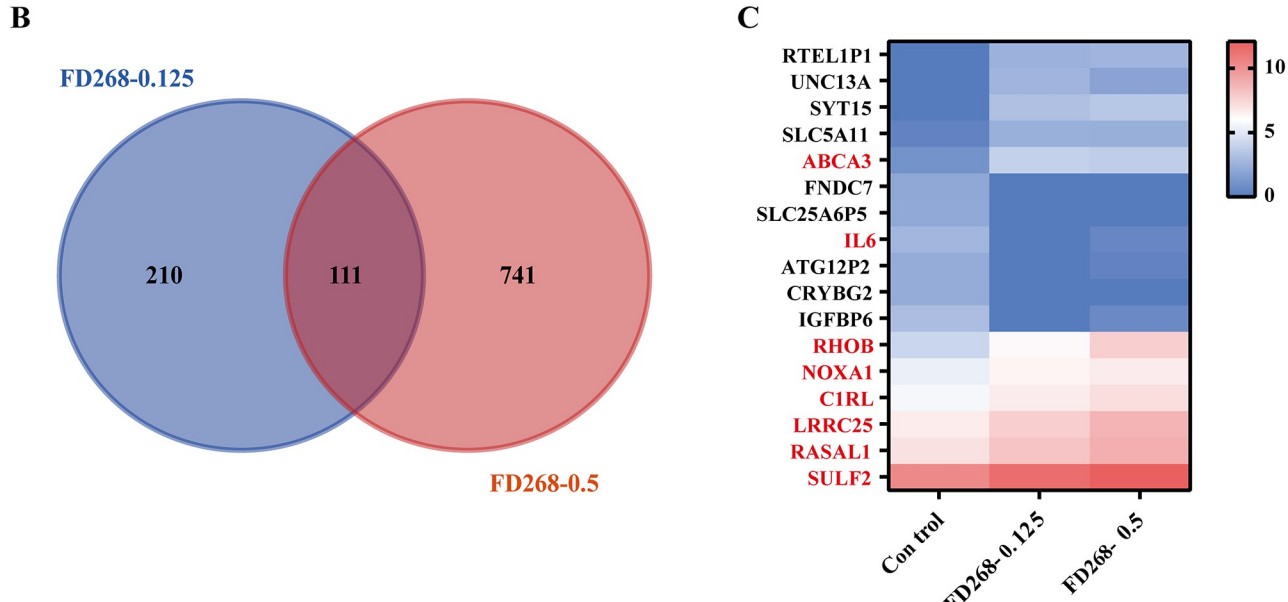

**Fig 8. FD268 regulates cell survival related genes and pathways in HL-60 cells.** (A) Kyoto Encyclopedia of Genes and Genomes (KEGG) pathway analysis of RNA-seq data in 0.125 μM FD268-treated group. The top six enriched pathways of common deferentially expressed genes are shown. The DEGs are involved in several metabolic pathways related to cell survival including PI3K-Akt signaling pathway and a set of main metabolism and signal transduction pathways was involved. (B) Overlapped deferentially expressed target genes of 0.125 μM FD268-treated and 0.5 μM FD268-treated group. (C) Expression profiles of overlapped common of target genes. The color in heatmap represents the normalized expression value.

examined and confirmed on these two cell lines respectively. Meanwhile, FD268 exerted the anti-proliferation through induction of cell cycle arrest at $G_0/G_1$ phase on HL-60 while at $G_2/M$ phase on MOLM-16. Furthermore, dose-dependent expression of apoptosis-related gene Bcl-2, Bad and Mcl-1 was confirmed on HL-60 while not on MOLM-16. Nevertheless, western-blot analyses revealed similar dose-dependent expression of pro- and antiapoptotic proteins on both HL-60 and MOLM-16. Although chemotherapy-induced differential transcriptional and protein expression as well as the cell-line-dependent regulation of cell cycle arrest are common, it raises the question whether they are strongly associated with the inhibition of PI3K/Akt/mTOR signaling pathway by FD268. Previous studies have revealed that Akt is a primary mediator in this pathway [44, 45]. Its activation promotes cell cycle progression by downregulation of cyclin D1, maintains cell survival through inactivation of Bad meanwhile increase of the expression of Bcl-2 protein, and promotes cell growth by phosphorylation of

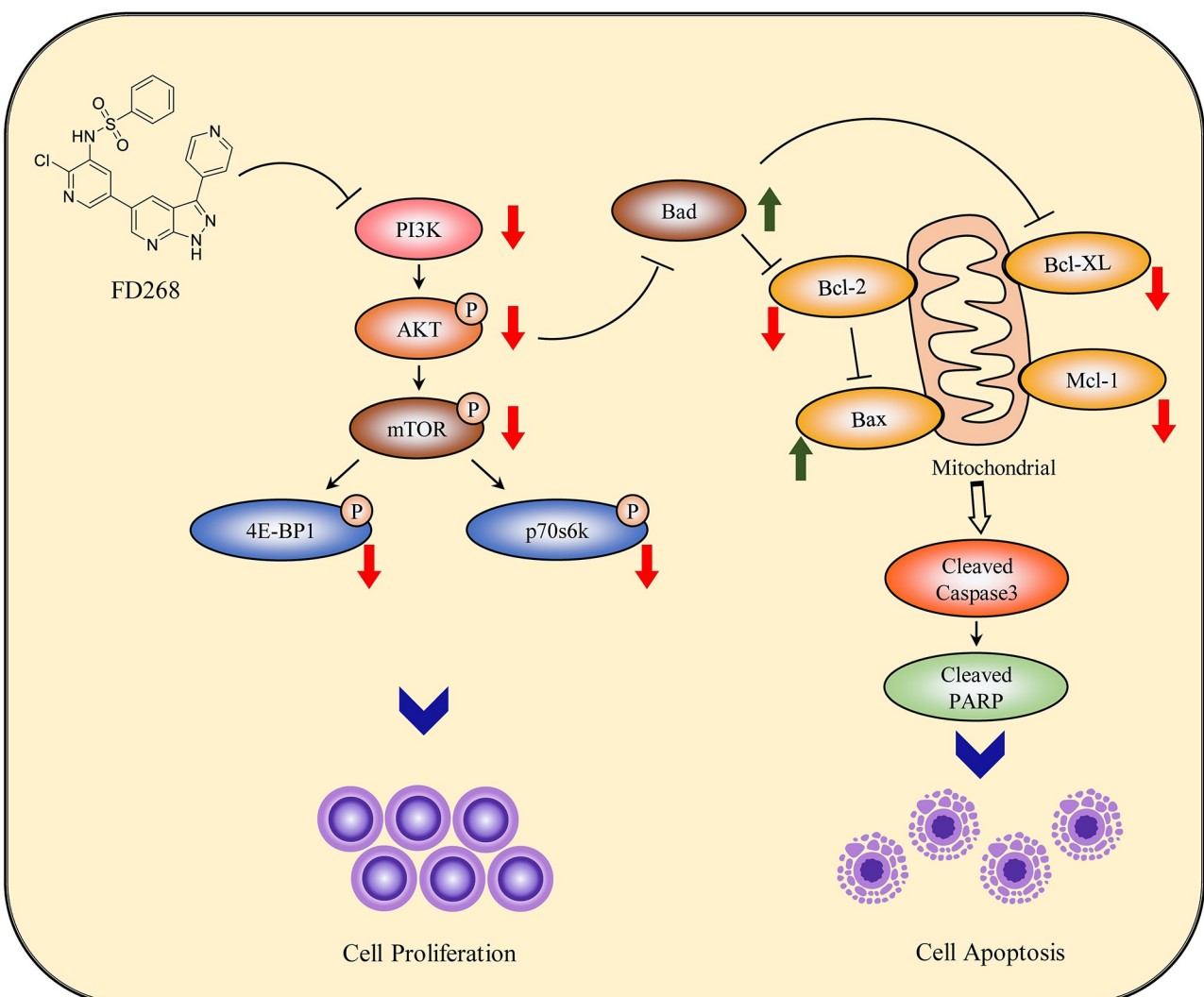

**Fig 9. Schematic for the plausible mechanisms by which PI3K pathway inhibition mediated the anti-leukemic activity of FD268 AML cells.** Via inhibition of the PI3K/Akt/mTOR signaling pathway, compound FD268 diminished the survival and proliferative capacity of the AML cells. The suppression of PI3K induced apoptotic cell death through upregulation of proapoptotic protein Bad and downregulation of antiapoptotic Mcl-1, and induced mitochondrial dependent cell apoptosis by activation of the caspase-3.

the downstream mTOR. Treated with FD268, our studies documented the reduction of PI3K/ Akt process and downregulation in the level of phosphorylated mTOR as well as its downstream targets p70S6K at Thr389 and 4EBP1 in both HL-60 and MOLM-16 cells, which is in agreement with those well-known PI3K inhibitors in treatment of AML [26, 46, 47]. Although it is hard to reveal the cell-line-dependent regulation of cell cycle arrest and completely map the mechanisms of actions in AML, our studies, together with KEGG pathway analysis, demonstrated the capacity of FD268 in suppressing cell proliferation and inducing apoptosis on AML through inhibition of PI3K/Akt/mTOR signaling pathway (Fig 9).

## 5 Conclusion

Our studies have documented for the first time that FD268, a pyridinesulfonamide derivative characterized by conjugation of 7-azaindole motif, possesses significant anti-leukemic activity towards AML cell lines by inhibition of PI3K/Akt/mTOR signaling pathway, thus supporting further development of pyridinesulfonamide derivatives as PI3K inhibitors for pre-clinical evaluation in treatment of AML.

## Supporting information

**S1 Fig. Morphological changes HL-60 and MOLM-16 treated with FD268 for 48h.** Cellular morphology of HL-60 and MOLM-16 treated with various concentrations of FD268 at 48 h. Phase contrast microscopy images and fluorescence microscopy images after staining with DAPI were observed. Cells incubated with DMSO (0.01%) were used control. Scale bar: 20 μm.
(TIF)

**S2 Fig. FD268 concentration-dependently inhibited the phosphorylation of AKT (Ser473) in HL-60 cells.** HL-60 cells were treated with indicated concentrations of FD268 for 24 h. The p-AKT(Ser473) expression and location was detected by immunofluorescence assay. The fluorescence microscopy images after staining with DAPI were observed. Scale bar: 20 μm.
(TIF)

**S3 Fig. Differentially expressed gene (DEGs) in HL-60 cells treated with FD268.** DESeq2 was used to screen differentially expressed genes among different groups ($|\log2FC| \geq 1$ and P value$\geq 0.05$). Volcano plot of deferentially expressed genes. (A) 0.125 μM FD268-treated group. (b) 0.5 μM FD268-treated group. 194 up-regulated and 184 down-regulated mRNAs were screened by comparison in group FD268-0.125μM, 655 up-regulated and 240 down-regulated mRNAs were screened by comparison in group FD268-0.5μM.
(TIF)

**S4 Fig. Gene Ontology (GO) analysis of RNA-seq data.** The top ten significantly enriched biological processes (BP), cell component (CC), molecule function (MF) are shown.
(TIF)

**S5 Fig. Expression level of overlapped common of target genes dose-dependently regulated by FD268.** HL-60 cells were treated by FD268 (0.125μM, 0.5 μM) for 24 h. The expression of target genes ABCA3, C1RL, LRRC25, NOXA1, RASAL1, AULF2 and IL-6 in HL-60 cells were determined using qRT-PCR analysis.
(TIF)

**S1 Table. The sequence of primers used for q-PCR.**
(XLSX)

**S2 Table. A total of terms annotated and categorized for the identified differential target genes.**
(XLSX)

**S3 Table. KEGG pathways annotated and categorized for the identified differential target genes.**
(XLSX)

**S4 Table. Target genes identified by overlapping the deferentially expressed of 0.125 μM FD268-treated and 0.5 μM FD268-treated group.**
(XLSX)

**S1 File. Raw images of Western blot from Figs 3–6.**
(DOCX)

**S1 Raw images.**
(PDF)

## Author Contributions

**Conceptualization:** Yi Chen, Yun Ling, Yaming Zhou.

**Formal analysis:** Yi Chen, Tianze Wu.

**Investigation:** Yi Chen.

**Methodology:** Yi Chen, Lei Lu.

**Project administration:** Mingli Deng, Yu Jia, Yongtai Yang, Xiaofeng Liu.

**Resources:** Chengbin Yang, Mingli Deng, Lei Lu.

**Supervision:** Yun Ling, Yaming Zhou.

**Validation:** Tianze Wu, Mingzhu Lu.

**Visualization:** Yi Chen.

**Writing – original draft:** Yi Chen.

**Writing – review & editing:** Zhenxia Chen, Hongyan Wang, Yun Ling, Lei Lu.

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
