## [Decision Letter · Decision Letter 0]

8 Aug 2022

PONE-D-22-00370

A Pyridinesulfonamide Derivative FD268 Suppresses Cell Proliferation and Induces Apoptosis via Inhibiting PI3K Pathway in Acute Myeloid Leukemia

PLOS ONE

Dear Dr. Zhou,

Thank you for submitting your manuscript to PLOS ONE. After careful consideration, we feel that it has merit but does not fully meet PLOS ONE’s publication criteria as it currently stands. Therefore, we invite you to submit a revised version of the manuscript that addresses the points raised during the review process by both Reviewers.

We look forward to receiving your revised manuscript.

Kind regards,

Francesco Bertolini, MD, PhD

Academic Editor

PLOS ONE

Journal Requirements:

2. Thank you for including your ethics statement:  "N/A".   

For studies reporting research involving human participants, PLOS ONE requires authors to confirm that this specific study was reviewed and approved by an institutional review board (ethics committee) before the study began. Please provide the specific name of the ethics committee/IRB that approved your study, or explain why you did not seek approval in this case.

Reviewers' comments:

Reviewer's Responses to Questions

**Comments to the Author**

1. Is the manuscript technically sound, and do the data support the conclusions?

Reviewer #1: Yes

Reviewer #2: Yes

2. Has the statistical analysis been performed appropriately and rigorously? 

Reviewer #1: Yes

Reviewer #2: Yes

3. Have the authors made all data underlying the findings in their manuscript fully available?

Reviewer #1: Yes

Reviewer #2: Yes

4. Is the manuscript presented in an intelligible fashion and written in standard English?

Reviewer #1: Yes

Reviewer #2: Yes

5. Review Comments to the Author

Reviewer #1: Congratulations on the excellent article. Some considerations:

- in keywords: you do not need to include those that are already mentioned in the title (pyridinesulfonamide; acute myeloid leukemia; Apoptosis)

- objectives are not well highlighted

- in some tests, in the text of materials and methods, it is not clear whether triplicates and 3 independent experiments were carried out, or only 3 independent experiments. They are only explained in the subtitles.

- in figure 4c the name of the cell lines was missing

- the work has been done with two cell types, but in RNAseq only one cell line was chosen, but there was no explanation for this.

Reviewer #2: The manuscript “A Pyridinesulfonamide Derivative FD268 Suppresses Cell Proliferation and Induces Apoptosis via Inhibiting PI3K Pathway in Acute Myeloid Leukemia” by Chen et al. presents the results on therapeutic activity of novel and highly active FD268 inhibitor of PI3K. The authors found that FD268 inhibits the viabilities of AML cells with the efficacies superior to other two PI3K inhibitors, PI-103 and CAL-101. They showed that FD268 suppresses cell proliferation, induces cell cycle arrest and apoptosis of AML cells through the inhibition of PI3K activity.

In my humble opinion, I think that this work will be of interest to a broad audience.

Overall, the manuscript is well written and experiments were properly designed and executed. However, some minor corrections and improvements are necessary prior to its acceptance.

Comments and Suggestions:

Line 90: repeating words “and and”.

Line 108: Should read “…and treated with FD268…”

Line 178: Should read “…plates and treated with FD268 (0.125, 0.5 µM) or DMSO for 24 h. Cells were harvested…”

Line 225: Should read “The values were shown as the…”

Lines 249-250: Fig. 3C: there is no dose dependent inhibition of cleavage of caspase 3 and PARP, but increase instead. The sentence should be corrected as “…proteins were consequently examined by western blot assay, which showed a dose dependent increase of both caspase-3 and PARP cleavage (Fig 3C).”.

Line 258: Should read “Statistical significance is represented with asterisks *P < 0.05 and ***P < 0.001.”, as there is no **P < 0.01. It might be also better to use white background for S phase data, as labels/asterisks are barely visible in Figure 2B.

Figure 4C: Indicate which blots represent HL-60 and MOLM-16 cells. Label for cell lines is missing.

Line 288: The authors should mention here that the protein levels of other markers of apoptosis, e.g. Bcl-2, Bax and Bcl-XL, did not change significantly in FD268-treated cells.

Figure 5C: Label of MOLM-16 has to be corrected (swapped letters). The same applies to Figure 6A and 6B.

Figure 9: Shouldn´t the cleaved caspase-3 and PARP lead to cell death/apoptosis, instead of cell survival (check the bottom right corner)?

6. PLOS authors have the option to publish the peer review history of their article (what does this mean?). If published, this will include your full peer review and any attached files.

Reviewer #1: No

Reviewer #2: No

---

## [Author Response · Author response to Decision Letter 0]

21 Sep 2022

Dear Editor and reviewers，

Thank you for your decision and constructive comments on our manuscript entitled “A Pyridinesulfonamide Derivative FD268 Suppresses Cell Proliferation and Induces Apoptosis via Inhibiting PI3K Pathway in Acute Myeloid Leukemia”. Those comments are all valuable and helpful for revising and improving our paper. We have studied all comments carefully and have made conscientious correction in the manuscript. Revised portion are marked in red and highlighted in this paper.

Revision notes, point-to-point, are given as follows:

Reviewer #1: 

Comment 1: in keywords: you do not need to include those that are already mentioned in the title (pyridinesulfonamide; acute myeloid leukemia; Apoptosis).

Response 1: Thank you for the suggestion about the key words. The key words that mentioned in the title has been replaced with the the following words: PI3K inhibitor, Haematological cancer and HL-60.

Comment 2: objectives are not well highlighted.

Response 2: Thank you for this valuable feedback. Our research have documented for the first time that a novel pyridinesulfonamide scaffold possesses significant anti-leukemic activity towards AML cells, the objective of our paper is about the early exploration of the mechanism of novel drug development. In the text of this paper, the objectives was not well highlighted due to the ways of language expression. We have further clarified this in the end of the Introduction section (lines 69-71), and the Conclusion section ( lines 457-461).

Comment 3: in some tests, in the text of materials and methods, it is not clear whether triplicates and 3 independent experiments were carried out, or only 3 independent experiments. They are only explained in the subtitles.

Response 3: Thank you for your questions. All the cells in each cell-based assay were seeded and harvested in triplicates, and at least three independent experiments were carried out in the CCK-8 cell proliferation assay and cell apoptosis assay. We have in added examinations in the text of materials and methods (lines 104-105, lines 120-121 and lines 145-146).

Comment 4: in figure 4c the name of the cell lines was missing.

Response 4: Thanks for pointing out about the missing details, the name of cell lines has been added in Figure 4C. 

Comment 5: the work has been done with two cell types, but in RNA-seq only one cell line was chosen, but there was no explanation for this.

Response 5: We appreciate for this question. Based on the previous studies of compound FD268, we found that FD268 has potential AML cell proliferation inhibitory activity, and further screening assay showed that the HL-60 cell line was more sensitive to compound FD268, so in the subsequent mechanistic explorations we chose HL-60 for the RNA-seq analysis. 

Reviewer #2:

Comment 1: Line 90: repeating words “and and”.

Comment 2: Line 108: Should read “…and treated with FD268…”

Comment 3: Line 178: Should read “…plates and treated with FD268 (0.125, 0.5 µM) or DMSO for 24 h. Cells were harvested…”

Comment 4: Line 225: Should read “The values were shown as the…”

Response 1-4: Thanks for the kind comments. The tense and repetition mistakes have been corrected in the manuscript.

Comment 5: Lines 249-250: Fig. 3C: there is no dose dependent inhibition of cleavage of caspase 3 and PARP, but increase instead. The sentence should be corrected as “…proteins were consequently examined by western blot assay, which showed a dose dependent increase of both caspase-3 and PARP cleavage (Fig 3C).”

Response 5: Thanks for the comments. In Figure. 3C, the western blot was applied to detect cleaved PARP, cleaved caspase-3 and caspase-3 as indicators of apoptotic cell death. The cleavage of caspase-3 and PARP increased dose dependently. We have changed the sentences as suggested. (Lines 249-250)

Comment 6: Line 258: Should read “Statistical significance is represented with asterisks *P < 0.05 and ***P < 0.001.”, as there is no **P < 0.01. It might be also better to use white background for S phase data, as labels/asterisks are barely visible in Figure 2B.

Response 6: Thanks for the valuable suggestion and we have removed the “**P < 0.01” in statistical significance annotation of Figure 2. The background for S phase data bar have been re-edited to highlight labels/asterisks.

Comment 7: Figure 4C: Indicate which blots represent HL-60 and MOLM-16 cells. Label for cell lines is missing.

Response 7: Thanks for pointing out about the missing, the name of cell lines has been added in Figure 4C. 

Comment 8: Line 288: The authors should mention here that the protein levels of other markers of apoptosis, e.g. Bcl-2, Bax and Bcl-XL, did not change significantly in FD268-treated cells.

Response 8: Thanks for this kind suggestion. At protein level, FD268 increased the expression of Bad, meanwhile inhibited Mcl-1 in both HL-60 and MOLM-16 dose-dependently. While, the protein levels of other markers of apoptosis, Bcl-2, Bax and Bcl-XL, did not change significantly in FD268-treated cells. We have added a sentence to to clarify this. (Lines 291-292)

Comment 9: Figure 5C: Label of MOLM-16 has to be corrected (swapped letters). The same applies to Figure 6A and 6B.

Response 9: Thanks for pointing out the mistakes, the label of MOLM-16 have been corrected in Figure 5C, 6A and 6B.

Comment 10: Figure 9: Shouldn´t the cleaved caspase-3 and PARP lead to cell death/apoptosis, instead of cell survival (check the bottom right corner)?

Response 10: Thank you for this valuable comment. In this diagram, we meant to show that the compound FD268 induced apoptosis through the activation of caspase-3 and PARP cleavage, so the label “cell Survival” have been changed into “Cell Apoptosis”. 

After careful revision, we worked on the manuscript for a long time and fixed all of these format or grammar issues on the manuscript. Thanks again to the reviewers and editors for you hard work! 

Sincerely yours，

Prof. Yaming Zhou

---

## [Decision Letter · Decision Letter 1]

5 Oct 2022

PONE-D-22-00370R1A Pyridinesulfonamide Derivative FD268 Suppresses Cell Proliferation and Induces Apoptosis via Inhibiting PI3K Pathway in Acute Myeloid LeukemiaPLOS ONE

Dear Dr. Zhou,

Thank you for submitting your manuscript to PLOS ONE. After careful consideration, we feel that it has merit but does not fully meet PLOS ONE’s publication criteria as it currently stands. Therefore, we invite you to submit a revised version of the manuscript that addresses the points raised during the review process.

ACADEMIC EDITOR: Please insert comments here and delete this placeholder text when finished. Be sure to:Indicate which changes you require for acceptance versus which changes you recommendAddress any conflicts between the reviews so that it's clear which advice the authors should followProvide specific feedback from your evaluation of the manuscriptPlease ensure that your decision is justified on PLOS ONE’s publication criteria and not, for example, on novelty or perceived impact.

We look forward to receiving your revised manuscript.

Kind regards,

Jianhong Zhou

Support Staff - Editorial

PLOS ONE

Journal Requirements:

Additional Editor Comments (if provided):

Reviewers' comments:

Reviewer's Responses to Questions

**Comments to the Author**

1. If the authors have adequately addressed your comments raised in a previous round of review and you feel that this manuscript is now acceptable for publication, you may indicate that here to bypass the “Comments to the Author” section, enter your conflict of interest statement in the “Confidential to Editor” section, and submit your "Accept" recommendation.

Reviewer #1: All comments have been addressed

Reviewer #2: All comments have been addressed

2. Is the manuscript technically sound, and do the data support the conclusions?

Reviewer #1: Yes

Reviewer #2: Yes

3. Has the statistical analysis been performed appropriately and rigorously? 

Reviewer #1: Yes

Reviewer #2: Yes

4. Have the authors made all data underlying the findings in their manuscript fully available?

Reviewer #1: Yes

Reviewer #2: Yes

5. Is the manuscript presented in an intelligible fashion and written in standard English?

Reviewer #1: Yes

Reviewer #2: Yes

6. Review Comments to the Author

Reviewer #1: Now, The manuscript is much better! All notes have been answered.

The reading is better understood after all mistakes are solved.

Reviewer #2: The authors have revised the manuscript according to my suggestions and recommendations. I have no further comments.

7. PLOS authors have the option to publish the peer review history of their article (what does this mean?). If published, this will include your full peer review and any attached files.

Reviewer #1: No

Reviewer #2: No

---

## [Author Response · Author response to Decision Letter 1]

3 Nov 2022

Dear reviewers,

Thank you for your constructive comments on our manuscript (ID: PONE-D-22-00370R2). Those comments are all valuable and helpful for revising and improving our paper. We have considered all comments carefully and made conscientious correction. Attached please find the revised manuscript entitled “A Pyridinesulfonamide Derivative FD268 Suppresses Cell Proliferation and Induces Apoptosis via Inhibiting PI3K Pathway in Acute Myeloid Leukemia” for your consideration. Point-to-point responses for the original manuscript were given in the file ‘Response to reviewers.docx’ and the correction were highlighted in the revised manuscript with track changes. There is no correction according to the recent comments for the R1 version. 

Thank you very much for your time! We would be glad to respond to any further questions and comments that you may have.

Sincerely yours

Prof. Yaming Zhou

---

## [Editor Report · Decision Letter 2]

6 Nov 2022

A Pyridinesulfonamide Derivative FD268 Suppresses Cell Proliferation and Induces Apoptosis via Inhibiting PI3K Pathway in Acute Myeloid Leukemia

PONE-D-22-00370R2

Dear Dr. Zhou,

We’re pleased to inform you that your manuscript has been judged scientifically suitable for publication and will be formally accepted for publication once it meets all outstanding technical requirements.

Kind regards,

Jianhong Zhou

Staff Editor

PLOS ONE
---

## [Editor Report · Acceptance letter]

10 Nov 2022

PONE-D-22-00370R2 

A Pyridinesulfonamide Derivative FD268 Suppresses Cell Proliferation and Induces Apoptosis via Inhibiting PI3K Pathway in Acute Myeloid Leukemia 

Dear Dr. Zhou:

I'm pleased to inform you that your manuscript has been deemed suitable for publication in PLOS ONE. Congratulations! Your manuscript is now with our production department. 

Kind regards, 

on behalf of

Dr. Francesco Bertolini 

Academic Editor

PLOS ONE